# The promise of open survey questions—The validation of text-based job satisfaction measures

Indy Wijngaards[1,2¤]*, Martijn Burger[1,3¤], Job van Exel[2¤]

**1** Erasmus Happiness Research Organization, Erasmus University Rotterdam, Rotterdam, the Netherlands, **2** Erasmus School of Health & Policy Management, Erasmus University Rotterdam, Rotterdam, the Netherlands, **3** Erasmus School of Economics and Tinbergen Institute, Erasmus University Rotterdam, Rotterdam, the Netherlands

¤ Current address: Erasmus University Rotterdam, Rotterdam, The Netherlands
* wijngaards@ese.eur.nl

**Data Availability Statement:** All relevant data are within the manuscript and its Supporting Information files.

**Funding:** IW received support from the Netherlands Organisation for Scientific Research

## Abstract

Recent advances in computer-aided text analysis (CATA) have allowed organizational scientists to construct reliable and convenient measures from open texts. As yet, there is a lack of research into using CATA to analyze responses to open survey questions and constructing text-based measures of psychological constructs. In our study, we demonstrated the potential of CATA methods for the construction of text-based job satisfaction measures based on responses to a completely open and semi-open question. To do this, we employed three sentiment analysis techniques: Linguistic Inquiry and Word Count 2015, SentimentR and SentiStrength, and quantified the forms of measurement error they introduced: specific factor error, algorithm error and transient error. We conducted an initial test of the text-based measures' validity, assessing their convergence with closed-question job satisfaction measures. We adopted a time-lagged survey design ($N_{wave\ 1}$ = 996; $N_{wave\ 2}$ = 116) to test our hypotheses. In line with our hypotheses, we found that specific factor error is higher in the open question text-based measure than in the semi-open question text-based measure. As expected, algorithm error was substantial for both the open and semi-open question text-based measures. Transient error in the text-based measures was higher than expected, as it generally exceeded the transient error in the human-coded and the closed job satisfaction question measures. Our initial test of convergent and discriminant validity indicated that the semi-open question text-based measure is especially suitable for measuring job satisfaction. Our article ends with a discussion of limitations and an agenda for future research.

## Introduction

Organizational scientists have been leveraging written texts in their study of psychological constructs and business phenomena for decades [1–3]. While manual human coding of texts continues to be the gold standard for annotating text, rapid advances in computer-aided text analysis (CATA) have opened up a venue for analyzing open texts in a drastically more

(NWO), grant number 652.001.003. See: https://www.nwo.nl/. The funders had no role in study design, data collection and analysis, decision to publish, or preparation of the manuscript.

**Competing interests:** The authors have declared that no competing interests exist.

efficient but still reliable manner [4]. CATA is a kind of content analysis that facilitates the measurement of constructs by converting text into quantitative data based on word frequencies [4]. One of the most popular applications of CATA is sentiment analysis, the practice of automatically detecting opinions, sentiments, attitudes and emotions about certain objects in human-generated texts [5,6]. Social scientists typically perform computer-aided sentiment analysis on bodies of relatively lengthy texts [e.g., 7,8], probably because detecting sentiment in short informal texts is more challenging than sentiment detection in lengthier texts [9,10]. It is therefore not surprising that research on performing CATA on brief responses to open survey questions thin on the ground within the organizational sciences [11]. This is unfortunate, as surveys still have a prominent place in organizational scientists' methodological toolboxes [12,13], open survey questions can function as valuable supplements to their closed counterparts [14–16], and researchers outside the discipline have proposed a multitude of promising solutions for mining textual survey data [17–22].

Complementing closed questions with open questions may have various benefits. First, open questions can function as a counter to the common method biases in questionnaires that primarily contain closed questions [23]. For instance, closed question survey scales can suffer from careless responding [24]. We believe the inclusion of open questions could be used to get respondents to respond more carefully, as open questions force respondents into a different and possibly more intensive form of cognitive processing [25]. Second, complementing closed questions with open questions facilitates triangulation of methods [26]. Next to using open questions to determine the construct validity of closed questions (and vice versa), researchers could leverage the responses to open questions to obtain a more holistic perspective on the construct of study [26,27]. Researchers could, for example, use the responses to assess when, why and how a construct is manifested [16] and unravel the psychological processes that influence the self-report responses to closed survey questions [28], because open questions naturally prompt more spontaneous and elaborate responses [29].

Job satisfaction is a construct that organizational researchers typically study with closed questions [for examples, see 30], and it has gained considerable attention in the literature [31]. Job satisfaction has rarely been studied using text-based measures based on responses to open questions [11]. We argue that this is a missed opportunity, as evidence suggests that the measurement of job satisfaction is complex [32] and that closed question job satisfaction measures tend to suffer from careless responding [33]. Sentiment analysis appears to be a suitable method for the creation of a text-based measure. As suggested by several empirical studies [e.g., 34–38], the sentiment found in texts seems to be a natural manifestation of the pleasant and unpleasant emotions, beliefs and cognitions employees have–factors that jointly constitute job satisfaction [32, e.g., 39,40].

We address two issues in this paper to illustrate the promise of open job satisfaction questions: (1) we investigate the reliability of computer-aided sentiment analysis for constructing a text-based job satisfaction measures, and (2) carry out an initial test of the measures' construct validity. We focus on two open-ended response formats: a substitution open job satisfaction question (called hereafter: open question) and a substitution semi-open job satisfaction question (called hereafter: semi-open question). The two questions, "How do you think about your job as a whole?", and "What three to five adjectives come to mind when you think of your job?", are similar in the sense that both are designed to measure general job satisfaction, and are intended as equivalents of closed survey questions [cf. 41]. The questions primarily differ in the degree to which they stimulate respondents to generate structured textual responses. Open questions allow respondents to decide on the length of their textual response, while semi-open questions are much more constraining.

While to date semi-open questions have rarely been used to measure attitudes [42], they may have several advantages over open questions. Respondents may prefer semi-open questions, because it takes less time and effort to write down several words that readily come to mind than to write down elaborate sentences (for some descriptive statistics about response burden of different question types, see [43]). In addition, researchers interested in quantitatively measuring constructs may favor semi-open questions over open questions, as responses to semi-open questions are more convenient for computer-aided sentiment analysis methods to analyze. Even though structured responses to semi-open questions are inherently short, these responses are likely to contain a higher proportion of useful and easy-to-process text than the unstructured texts that open questions generate. Structured texts are likely to contain limited syntax and mainly useful, emotion-loaded words, e.g., adjectives generally carry subjective content [44], whereas unstructured informal texts typically contain a high proportion of irrelevant words, e.g., articles, conjunctions, typing mistakes and negations, which are difficult to deal with using computer-aided sentiment analysis methods [10].

We contribute to the survey methodology literature by addressing the methodological dilemma of choosing between human and computer-aided content analysis. Even though human coding remains the gold standard for content analysis [4], e.g., sentiment detection in text [45], it can be very time-consuming and expensive [15] and therefore sometimes unfeasible [34]. CATA methods are considerably more efficient and can save resources, but inevitably produce measures with attenuated reliability [46]. To assist organizational researchers that face this trade-off, we systematically assessed the degree of measurement error in text-based measures by following the guidelines set by McKenny, Aguinis, Short and Anglin [46]. In particular, we studied the three sources of measurement error that are relevant for CATA research: specific factor error, algorithm error and transient error [46]. Additionally, we are among the first to use CATA to construct text-based job satisfaction measures from responses to an open question (hereafter called: open text-based measure) and a semi-open question (hereafter called: semi-open text-based measure) and to test their convergent and discriminant validity by means of closed question measures.

This paper is structured as follows. In the remainder of this section, we discuss the three computer-aided sentiment analysis techniques that were used in this study and formulate hypotheses. Next, we describe our sample, procedures and analytical strategy. We then present our comparative analysis and test our hypotheses. Finally, we discuss our findings and provide an agenda for future research.

## Sentiment analysis approaches

Manual and computer-aided sentiment analysis can be used to construct a measure from the responses to open questions [14,34]. Human coders subjectively rate text in terms of sentiment. To date, ratings by human coders have been treated as the gold standard and benchmark for computer-aided sentiment analysis [10,44,45]. However, when large volumes of texts have to be analyzed, complete reliance on human coders to analyse texts manually may become infeasible [46]. Moreover, humans may introduce bias in sentiment ratings [14,47], e.g., through individual differences in evaluation strategies [48] and in annotation experience and education [49]. To mitigate these biases, researchers often make sure that multiple individuals independently annotate the same texts, calculate an inter-rater reliability score and compute an average rater score [36,38,48].

Two general streams of methods can be identified in computer-aided sentiment analysis: lexicon-based (or CATA-based) and learning-based techniques [44]. Lexicon-based methods involve the detection of sentiment in texts based on a dictionary of words that are labeled to

reflect their semantic orientation, i.e., polarity and strength of words. Learning-based methods make use of labeled instances of text to build classifiers. Put differently, these methods use textual data that are already labeled with their semantic orientation to train an algorithm, with the purpose of predicting (i.e. classifying) unlabeled textual instances.

Both methods have certain advantages and disadvantages. Learning-based techniques often perform well in the domain that they have been trained, but lack accuracy when training data are small or the classifier is used in another domain [44,50]. Lexicon-based methods do not suffer from these problems, as they do not rely on training data [50], and domain-specific words can be added to a general dictionary to make a dictionary perform well in specific contexts [44]. However, their accuracy typically drops when textual data contains semantic rules and linguistic nuances like sarcasm [50].

In this study, we relied on lexicon-based methods, as no appropriate dataset was available to train a learning-based sentiment analysis algorithm. We employed Linguistic Inquiry and Word Count (LIWC) 2015 [51], SentiStrength [10] and SentimentR [52] to construct the text-based measures. We describe the similarities and differences between the software programs below.

**LIWC 2015.** The LIWC software is arguably the most widely used CATA technique in the organizational sciences [3], and has been systematically validated in a large number of studies [53]. The LIWC software generates scores on a wide variety of constructs, e.g., social orientation, honesty, affective tone, by looking up words in an English dictionary of 6,400 words, word stems, and emoticons. Turning to sentiment analysis, the LIWC 2015 software includes a transparent dictionary for positive emotion (620 words) and negative emotion (744 words) [53], and contains a commercially licensed, non-transparent emotional tone variable [54] that summarizes the positive and negative emotion variables into one sentiment score [53].

**SentimentR.** SentimentR is a sentiment analysis software package that is freely available on CRAN [55]. Various studies outside the organizational sciences have successfully adopted this software for sentiment classification tasks [e.g., 56,57], and proved its superior performance to other software programs, such as LIWC [52,58,59]. By default, SentimentR uses Jockers' [60] English dictionary which contains 10,739 words [52]. Besides being open-access, non-commercialized and specially designed for sentiment analysis, it differs from LIWC in one major way. SentimentR does not just count individual words; the algorithm considers valence shifters to improve the accuracy of its semantic polarity recognition. Valence shifters can be split into negators, amplifiers, deamplifiers, and adversative clauses. A negator changes the sign of a sentiment-loaded word (e.g., 'I do *not* enjoy my work'); an amplifier enhances the impact of a sentiment-loaded word on the overall sentiment score (e.g., 'I *truly* enjoy my job'); a deamplifier reduces the impact (e.g., 'I *hardly* enjoy my work'); an adversative conjunction overrides a sentiment-loaded clause (e.g., 'I enjoy my work, *but* hate my boss'). In SentimentR, the valence shifters are considered by weighting the valence shifters found four words before and two words after the polarized word.

**SentiStrength.** Just like SentimentR, SentiStrength is a software program that is specially designed to detect sentiment in texts and is freely available for non-commercial users [10]. It is optimized for sentiment analysis of short informal texts [10, e.g., tweets, 61], and has found to be reliable [62–64]. The SentiStrength dictionary is constructed from several well-validated, English dictionaries–LIWC 2003 [65] and the General Inquirer [66]–and contains 2,310 words [67]. The developers of SentiStrength deployed a learning-based sentiment analysis technique to optimize the software's performance. As a consequence, the software considers textual aspects such as punctuation (e.g., exclamation marks), the use of multiple vowels (e.g., *haaappy*), frequently used idioms (e.g., *I am like you* and *I like you*) and valence shifters in its sentiment score calculation.

## Measurement error in text-based measures

In general, three categories of CATA measurement error exist: specific factor error, algorithm error and transient error [46]. Specific factor error relates to the word lists that the CATA method uses, and the extent to which they are fit for the task at hand. Specific factor error can be assessed by computing the parallel forms reliability. In our case, this meant examining the convergence between computer-generated and human-coded sentiment measures. Algorithm error is related to the extent to which the measures produced by different CATA techniques vary. The more the measures diverge, the higher the algorithm error will be. This can be thought of as 'interalgorithm' error and can be assessed with by Krippendorff's alpha ($\alpha$) inter-rater agreement estimate [2,68]. Transient error is caused by the temporal factors that can impact the responses by a respondent. For instance, mood states can affect the overall valence in a textual response. Transient error can be measured by calculating test-retest reliability.

**Specific factor error.** Open questions inevitably produce more unstructured texts than semi-open questions, because of the absence of answering constraints. These texts are likely to contain a high proportion of non-emotion loaded words, misspellings, semantic rules and valence shifters and linguistic characteristics that are generally difficult for lexicon-based sentiment analysis to process [10]. As humans are typically the most competent to detect sentiment from natural, unstructured texts [67] and individual adjectives that the semi-open question produces are relatively straight-forward to look-up in sentiment dictionaries, we hypothesized that:

*Hypothesis 1a*. The specific factor error in open text-based measures will be higher than the specific factor error in semi-open text-based measures.

As the selected software packages vary in their suitability to analyze short informal texts, we predicted that the specific factor error in open text-based measure varies from one software package to the other. We expected that SentiStrength will have the highest accuracy, because it was designed for the sentiment analysis of short informal texts, has the most advanced algorithm of the three and has outperformed LIWC 2007 in the analysis of short texts [63,69]. Further, we predicted that SentimentR would produce more reliable measures than LIWC 2015, because, contrary to LIWC 2015, SentimentR considers valence shifters in its algorithm. As such, we hypothesized that:

*Hypothesis 1b*. The specific factor error will vary across open text-based measures, with LIWC 2015 performing the worst and SentiStrength performing the best.

Semi-open questions only generate context-free words that CATA methods can conveniently look up in their dictionaries without any substantial pre-processing, e.g., stemming and removing stop words. For this reason, the accuracy of a semi-open text-based measure almost exclusively depends on the quality and completeness of the dictionary. We deemed two competing hypotheses plausible. One the one hand, we could expect that SentiStrength and LIWC 2015 will outperform SentimentR, because the LIWC dictionary is more systematically validated than Jockers' dictionary. On the other hand, we could expect the SentimentR measure to be most reliable, since the Jockers' dictionary is at least four times bigger than the dictionaries of the other software programs and its overall word coverage is the highest of the methods discussed here. We expected that these benefits cancel each other out, and therefore predicted that:

*Hypothesis 1c*. The specific factor error in semi-open text-based measures will not vary across software packages.

**Algorithm error.**   The LIWC 2015, SentimentR and SentiStrength software all use different dictionaries and algorithms for their sentiment analysis, which inevitably causes their measures to vary. The respective dictionaries varied in size, the sentiment coding schemes differed, and the algorithms diverged in their capability to control for semantic rules and nuances. As such, we expected that the agreement between algorithms would not exceed the lower bound for acceptable agreement for human coders, i.e., 80% [70].

*Hypothesis 2.* Substantial algorithm error exists between LIWC 2015, SentimentR and

SentiStrength measures, as demonstrated by an average agreement of lower than 80%.

**Transient error.**   We did not expect complete consistency of language over time, as demonstrated in very high test-retest reliabilities, because the open questions were designed to measure job satisfaction. Test-retest reliability of job satisfaction measures after one year is typically below .6 and above .2 [71]. For this reason, we moved beyond an assessment of absolute test-retest reliability and examined the relative test-retest reliability of the text-based measures. We did this by comparing the text-based measures' test-retest reliabilities with the test-retest reliabilities of the human-coded measures and closed question job satisfaction measure. We hypothesized that:

*Hypothesis 3a.* The test-retest reliability of the text-based measures will deviate less than .2 from the test-retest reliability of the human-coded measures.

*Hypothesis 3b.* The test-retest reliability of the text-based measures will deviate less than .2 from the test-retest reliability of the closed question job satisfaction measure.

**Validity as measure of job satisfaction.**   The open and semi-open question from this study were designed to measure general job satisfaction. This is why, it is pivotal to assess the construct validity of the measures. We did this by examining their convergent and discriminant validity.

First, we tested the measures' convergent validity, the extent to which the two measures that purport to measure the same construct show strong empirical agreement. The few studies that linked text-based measures to closed job satisfaction measures found moderate correlations (e.g., 11,34,36,38). Hence, we predicted that the text-based measures and closed question measures of both general job satisfaction and measures of job facet satisfaction would converge. In addition, we expected that the correlations between the text-based measures and the general job satisfaction measure would be higher than the ones between the text-based measures and the job facet satisfaction measures. We therefore hypothesized:

*Hypothesis 4a.* The open and semi-open text-based measures will converge with closed question measures of job satisfaction, as demonstrated in positive, significant correlations.

*Hypothesis 4b.* The open and semi-open text-based measure will converge more strongly with the closed question measure of general job satisfaction than with the measures individual job facet satisfaction.

Discriminant validity, the degree to which the measures correspond to measures of related but distinct constructs, was assessed by comparing the correlations between the text-based measures and the closed question measure of general job satisfaction with the correlations

between the text-based measures and two antecedents, i.e., person-organization (P-O) fit and virtuous leadership, and three outcomes, i.e., life satisfaction, flourishing and organizational citizenship behavior (OCB).

P-O fit, "the compatibility between people and organisations that occurs when: (a) at least one entity provides what the other needs, or (b) they share similar fundamental characteristics, or (c) both" [72] is likely to contribute to job satisfaction, as feelings of fit spark feelings of need fulfilment. Virtuous leadership is a positive leadership characterized by six cardinal virtues: courage, temperance, justice, prudence, humanity and truthfulness [73,74]. It is a likely determinant of job satisfaction, as virtuous leadership behaviours are likely to be ethical [74] and ethical leadership positively affects job satisfaction [75]. We further predicted that job satisfaction is positively related to context-free well-being constructs, such as life satisfaction "the global assessment of a person's quality of life according to his own criteria" [76] and flourishing, a multi-dimensional construct that concerns "important aspects of human functioning ranging from positive relationships, to feelings of competence, to having meaning and purpose in life" [77], as domain-specific well-being tends to spill over into context-free well-being, and vice versa [78]. Finally, we expected that job satisfaction will also be associated with OCB, "helpful, constructive gestures exhibited by organisation members and valued or appreciated by officials, but not related directly to individual productivity nor inherently in the enforceable requirements of the individuals role" [79], because employees feel that they have to reciprocate good treatment by the organization (e.g., having a careful leader and doing an interesting job).

*Hypothesis 5a*. The open and semi-open text-based measures correlate positively with closed question measures of P-O fit, virtuous leadership, life satisfaction, flourishing and OCB.

*Hypothesis 5b*. The open and semi-open text-based measures will converge more strongly with a closed question measures of general job satisfaction than with closed question measures of P-O fit, virtuous leadership, life satisfaction, flourishing and OCB.

In light of our hypotheses about the higher reliability of the semi-open text-based measures, and the importance of reliability for a measure's validity [80], we also hypothesized that:

*Hypothesis 6a*. The semi-open text-based measure will show better convergent validity than the open text-based measure.

*Hypothesis 6b*. The semi-open text-based measure will show better discriminant validity than the open text-based measure.

## Materials and methods

### Procedure and sample

As we desired to obtain input from a large variety of respondents, we outsourced the data collection to Prolific. Prolific is a virtual crowdsourcing platform where people can complete paid tasks, in similar manner to that of Amazon's Mechanical Turk. Prolific has been found to collect good quality data [81]. Qualtrics was used for survey administration.

We used a two-wave time-lagged survey design to test our hypotheses. The demographic characteristics of the respondents that participated in our study are presented in Table 1. Using Prolific's filtering system, we selected people who were in full-time or part-time employment and lived in either the United States or the United Kingdom. The first wave of data

collection in December 2017 resulted in 997 valid responses. In March 2019, we used Prolific again to collect survey data from 125 respondents that had participated in the 2017 survey. Of the initial sample, the majority of the respondents were female, 74.6%. Most of the respondents were in a relationship, 76.5%. The average age was 35.6, and 76.7% of the respondents had at least some college experience. The demographic characteristics of the respondents from the second wave generally corresponded to the characteristics of the initial sample. At the beginning of the questionnaire, respondents were asked to give informed consent to their data being used for this research. Participation was completely voluntary with anonymity guaranteed.

## Measures

Here, we describe the open question and closed question measures from this study. Note: we used the 'Force response' option in Qualtrics, so we did not have any non-response in the data. Following recommendations of Dunn, Baguley and Brunsden [82], we report Cronbach's α and McDonald's [83] ω as measures of internal consistency of the multiple-item scales. Internal consistency of all multiple-item scales was good, as values of α and ω consistently exceeded .7 [84]. A summary of all measures was presented in Table 2.

**Job satisfaction.** We used closed, open and semi-open questions to measure job satisfaction. Eight closed job satisfaction questions were asked. One measured general job satisfaction and read "How satisfied are you with your job?". Seven questions measured satisfaction with job facets, i.e. work content, work-life balance, supervisor, team, company, work environment and pay, all of which had the same format: "How satisfied are you with the following: [Your

**Table 1. Demographics of the wave 1 (N = 997) and wave 2 (N = 116).**

| | Wave 1 | | Wave 2 | |
|---|---|---|---|---|
| Characteristic | N | % | N | % |
| Age | | | | |
| Mean | 35.6 | | 39.6 | |
| Standard deviation | 9.8 | | 10.5 | |
| Gender | | | | |
| Female | 744 | 74.6 | 68 | 58.6 |
| Male | 253 | 25.4 | 48 | 41.4 |
| Marital status | | | | |
| Divorced | 45 | 4.5 | 9 | 7.8 |
| In a relationship | 317 | 31.8 | 29 | 25.0 |
| Married | 446 | 44.7 | 55 | 47.4 |
| Single | 185 | 18.6 | 23 | 19.8 |
| Widowed | 4 | 0.4 | 0 | 0.0 |
| Education | | | | |
| Less than high school | 8 | 0.8 | 1 | 0.9 |
| High school graduate | 139 | 13.9 | 13 | 11.2 |
| Professional degree | 87 | 8.7 | 4 | 3.5 |
| Some college | 258 | 25.9 | 23 | 19.8 |
| 2-year degree | 79 | 7.9 | 13 | 11.2 |
| 4-year degree | 278 | 27.9 | 40 | 34.5 |
| Master's degree | 126 | 12.6 | 20 | 17.2 |
| Doctorate | 22 | 2.2 | 2 | 1.7 |

*N* = sample size; % = percentage

**Table 2. Summary of measures.**

| Measure | Words<br>M/SD | Rating/scores<br>Mean | SD | α |
|---|---|---|---|---|
| Closed question | | | | |
| *Wave 1* | | | | |
| General job satisfaction | | 6.42 | 2.37 | |
| Satisfaction with work environment | | 6.61 | 2.25 | |
| Satisfaction with work content | | 6.56 | 2.26 | |
| Satisfaction with team | | 7.45 | 2.08 | |
| Satisfaction with supervisor | | 6.77 | 2.79 | |
| Satisfaction with work-life balance | | 6.45 | 2.49 | |
| Satisfaction with company | | 6.47 | 2.52 | |
| Satisfaction with pay | | 5.44 | 2.47 | |
| P-O fit | | 5.00 | 1.30 | .87 |
| Virtuous leadership | | 4.78 | 1.38 | .97 |
| Life satisfaction | | 6.80 | 1.85 | |
| Flourishing | | 5.38 | 0.95 | .91 |
| OCB | | 3.13 | 0.78 | .87 |
| *Wave 2* | | | | |
| General job satisfaction | | 5.99 | 2.91 | |
| Open question | | | | |
| *Wave 1* | 48.47/39.72 | | | |
| Independent coders | | 3.35 | 1.09 | |
| LIWC 2015 | | 4.21 | 1.34 | |
| SentimentR | | 2.81 | 0.62 | |
| SentiStrength | | 3.09 | 0.72 | |
| *Wave 2* | 64.99/42.12 | | | |
| Independent coders | | 3.24 | 1.30 | |
| LIWC 2015 | | 3.77 | 1.55 | |
| SentimentR | | 3.09 | 0.83 | |
| SentiStrength | | 3.16 | 0.92 | |
| Semi-open question | | | | |
| *Wave 1* | 4.65/1.93 | | | |
| Independent coders | | 3.16 | 0.91 | .92 |
| LIWC 2015 | | 3.69 | 1.73 | .49 |
| SentimentR | | 3.22 | 0.76 | .74 |
| SentiStrength | | 3.41 | 0.68 | .91 |
| *Wave 2* | 4.45/1.62 | | | |
| Independent coders | | 3.00 | 1.30 | .87 |
| LIWC 2015 | | 3.16 | 1.87 | .69 |
| SentimentR | | 3.05 | 1.07 | .69 |
| SentiStrength | | 3.26 | 0.78 | .92 |

LIWC = Linguistic Inquiry and Word Count; M = Mean; SD = Standard deviation; α = Cronbach's α.

salary]?". Answer categories ranged on an 11-point scale from 0 (*very unsatisfied*) to 10 (*very satisfied*).

The open question we used, reads: "How do you think about your job as a whole?" We included an extra encouragement and three sub-questions to stimulate respondents to provide

a sufficiently elaborate answer, i.e., "It is of vital importance for our research that you take your time to provide a concise and complete answer to this question. Ask yourself questions like: 'How do I feel when I am working?', 'Am I happy with my job?' and 'Do I like my job?'". As another safeguard, we included a response validation of 20 or more characters. The mean number of words in wave 1 was 48. The mean number of words from respondents that completed both surveys was 54 in wave 1 and 65 in wave 2.

Concerning the semi-open question, we followed the guidelines of the ajective generation technique [85] to construct the following question: "Which three to five adjectives come to mind when you think of your job as a whole? Adjective 1: [. . .]–Adjective 5 [. . .]" Respondents were forced to report at least thee adjectives. The mean number of words in wave 1 was 4.7. The mean number of words from respondents that participated in both surveys was 4.6 in wave 1 and 4.5 in wave 2. It seems that most respondents were able to adhere to the answering constraints, as 56.5% of all words provided in wave 1 and 60.8% of all words provided in wave 2 were adjectives. To prepare the textual data for computer-automated sentiment analysis, we first performed a manual spelling check in Microsoft Excel 2016. Next, we omitted all non-alphabetic characters, e.g., punctuation, special characters and empty lines, and converted the texts into lowercase.

We illustrated the responses to the different job satisfaction questions by listing the ten most frequently used words for respondents who were dissatisfied with their job (general job satisfaction ≤ 4), neither dissatisfied nor satisfied (general job satisfaction = 5 or general job satisfaction = 6) and satisfied individuals (general job satisfaction ≥ 7) in Table 3. Several insights can be gained from these frequency tables. First, we see that most respondents were at least moderately satisfied with their job. Second, the results suggest that the most frequently used words in the responses to the semi-open questions corresponded well with the job satisfaction scores, while the responses to the open question are less straightforward to interpret. For instance, words such as 'job', 'work', 'feel' and 'like' can be found in the frequency tables of both the satisfied and dissatisfied respondents. The dissatisfied respondents use these words often together with a valence shifter (e.g., 'I do *not* like my work'). Third, we noticed that various seemingly negative adjectives, e.g., challenging, busy, stressful, are not only used by dissatisfied respondents.

**Table 3. Most frequently used words in responses to open and semi-open job satisfaction question.**

| Low job satisfaction (N = 193) | | | | Moderate job satisfaction (N = 242) | | | | High job satisfaction (N = 562) | | | |
|---|---|---|---|---|---|---|---|---|---|---|---|
| Open | | Semi-open | | Open | | Semi-open | | Open | | Semi-open | |
| N | Word* | N | Word | N | Word* | N | Word* | N | Word* | N | Word* |
| 244 | Job | 56 | Boring | 282 | Job | 39 | Rewarding | 650 | Job | 131 | Rewarding |
| 207 | Work | 41 | Stressful | 253 | Work | 35 | Challenging | 546 | Work | 103 | Challenging |
| 111 | Feel | 29 | Repetitive | 133 | Feel | 31 | Stressful | 326 | Feel | 97 | Interesting |
| 109 | Like | 25 | Tiring | 105 | Like | 30 | Busy | 248 | Happy | 79 | Busy |
| 53 | Enjoy | 19 | Busy | 83 | Enjoy | 27 | Interesting | 245 | Like | 73 | Fun |
| 53 | Get | 19 | Frustrating | 67 | Get | 26 | Boring | 224 | Enjoy | 50 | Important |
| 51 | Time | 16 | Challenging | 65 | Happy | 25 | Hard | 141 | Working | 44 | Stressful |
| 50 | People | 16 | Hard | 64 | People | 25 | Tiring | 131 | Can | 42 | Happy |
| 38 | Much | 13 | Rewarding | 58 | Can | 23 | Repetitive | 118 | People | 41 | Enjoyable |
| 37 | However | 13 | Dull | 49 | Time | 19 | Easy | 106 | Get | 40 | Exciting |

N = Number of observations.

* = The most frequently used stop words in the English language are omitted from the textual data [86].

**Virtuous leadership.** Virtuous leadership was measured using the 18-item Virtuous Leadership Questionnaire developed by Wang and Hackett [73] ($\alpha$ = .97; $\omega$ = .97). Answer categories ranged on a 7-point Likert scale from 1 (*never*) to 7 (*always*). An example question is "My supervisor expresses concern for the misfortunes of others".

**P-O fit.** P-O fit was measured using a 3-item scale developed by Cable and Judge [87] ($\alpha$ = .87; $\omega$ = .89). Answer categories ranged from 1 (*strongly disagree*) to 7 (*strongly agree*). An example item is "my values match those of current employees in my organisation."

**OCB.** OCB was measured using the 10-item short version of the Organisational Citizenship Behavior Checklist developed by Spector, Bauer and Fox [88] ($\alpha$ = .87; $\omega$ = .87). Response categories ranged from 1 (*never*) to 5 (*every day*). An example item is "How often have you lent a compassionate ear when someone had a work problem.".

**Flourishing.** Flourishing was measured using the 8-item Flourishing Scale developed by Diener and colleagues [77] ($\alpha$ = .91; $\omega$ = .91). Answer categories ranged from 1 (*strongly disagree*) to 7 (*strongly agree*). An example item is "I am optimistic about my future.".

**Life satisfaction.** Life satisfaction was measured with a single item that read "All things considered, how satisfied are you with your life as a whole these days?". The question stems from the World Values Survey [89], one of the largest and most comprehensive surveys that administers well-being questions across nations [90]. Answer categories ranged from 0 (*not satisfied at all*) to 10 (*very satisfied*).

## Analytical strategy

Data pre-processing and hypothesis testing was done in R [55]. For reproducibility purposes, all scripts (S1 File) and data (S1 Dataset) will be made available in the supplementary information.

**Sentiment detection.** We used both independent manual coding by humans and computer-aided coding, i.e., LIWC 2015, SentimentR and SentiStrength. Summaries of the textual responses and descriptive statistics of the text-based measures can be found in Table 2. The histograms of the different ratings from wave 1 are displayed in Fig 1 and Fig 2. To be able to make fair comparisons between text-based measures, we recoded or rounded sentiment scores into a categorical five-point scale: 1 (*very negative*), 2 (*negative*), 3 (*neutral*), 4 (*positive*) and 5 (*very positive*).

**Human coding.** Three coders were asked to independently rate all textual responses in terms of sentiment on a categorical five-point scale ranging from 1 (*very negative*) to 5 (*very positive*). Coders were asked to annotate all adjectives separately. The average time for coding 100 responses to the semi-open and open question was 120 minutes. To code all textual responses (about 1,100), coders spend 3,960 minutes (120 minutes * 3 coders * 11). Coders were provided a detailed guideline to ensure rater consistency (S1 Protocol). Following the recommendations of Hallgren [91], we used a two-way model, average-measures unit interclass correlation to determine inter-rater reliability. We deemed an interclass correlation of .6 to be minimally acceptable [92]. For waves 1 and 2, respectively, the interclass correlation scores for the first adjective (.994 and .920), second adjective (.902 and .914), third adjective (.904 and .936), fourth adjective (.817 and .904), fifth adjective (.903 and .916), complete semi-open text-based measure (.934 and .936) and open text-based measure (.921 and .951) exceeded this threshold. The human coding measure of the semi-open question was created by first averaging the sentiment ratings of the individual adjectives provided by the individual independent coders, and then taking the mean of the aggregated sentiment ratings. The human coding measure of the open question was generated by aggregating the sentiment ratings provided by the independent coders.

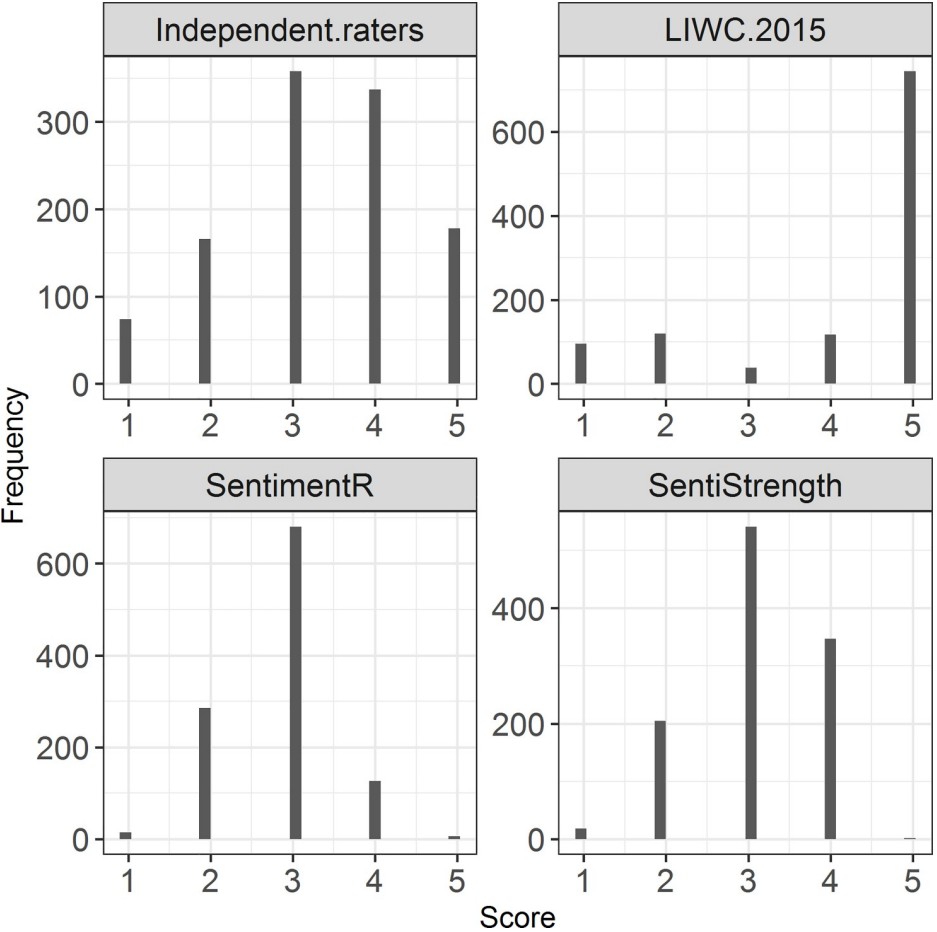

**Fig 1. Histograms of the sentiment measures based on the open question.**

We verified the reliability of this independent coding procedure by correlating the generated measures with the evaluations from respondents themselves. We asked respondents the following question after they completed the open questions: "How would you rate your previous answer in terms of sentiment/emotion?". The answer categories ranged from 1 (*very negative*) to 5 (*very positive*). The results showed that the semi-open measure based on independent coding correlated strongly with the respondent-generated semi-open measure ($r = .794$). We found the same pattern for the open text-based measures ($r = .785$).

**Computer-aided coding.** The LIWC 2015 measure originally ranged on a continuous scale from 1, extremely negative emotional tone, to 100, extremely positive emotional tone. SentimentR produced sentiment scores from -2 to 2. SentiStrength was not programmed to produce overall sentiment scores but was instead designed to generate scores for negative sentiment, range from -1 to -5, and positive sentiment, range from 1 to 5. The overall sentiment score was created by summing up the positive and negative score.

**Hypothesis testing.** The guidelines provided by McKenny et al. [46] were followed to assess measurement error in text-based measures. Pearson correlation analyses were used to assess specific factor error, i.e., examining the convergence between the individual text-based measures and the human ratings. The data from wave 1 and wave 2 were combined, because the software packages analyze texts as independent observations. To test hypothesis 1a, we first

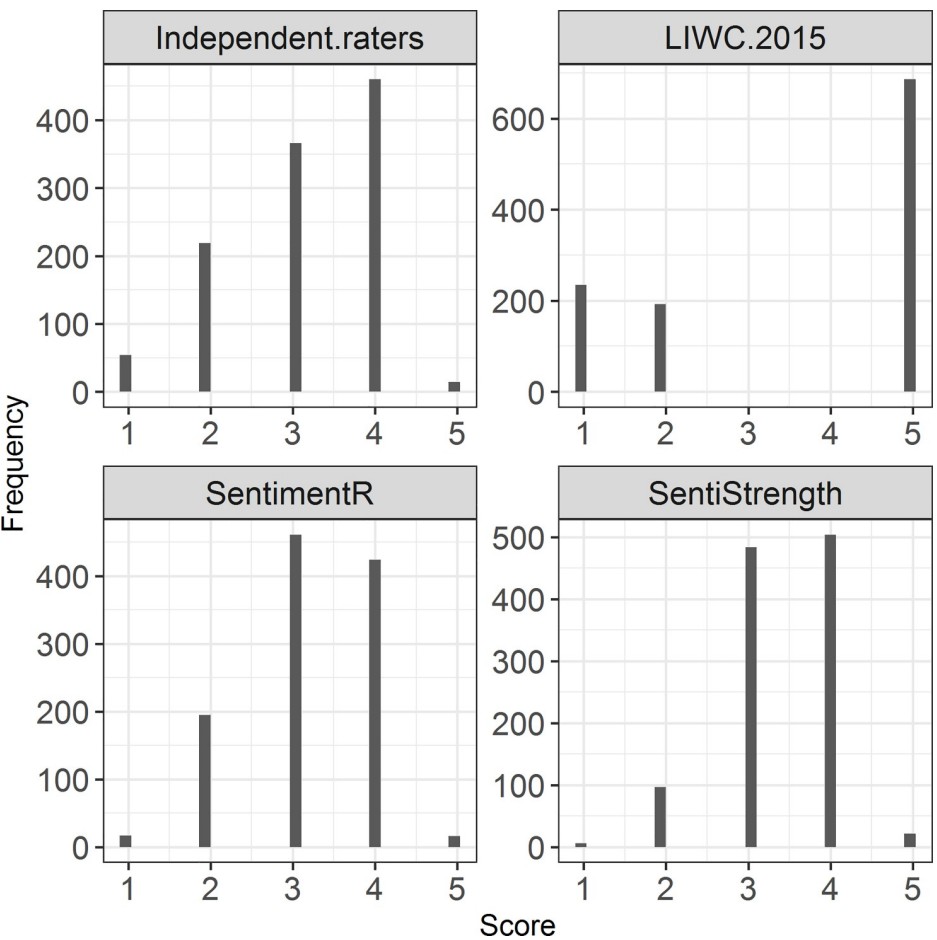

**Fig 2. Histograms of the sentiment measures based on the semi-open question.**

conducted a Fisher [93] $z$-transformation of all correlations between the individual text-based measures and the human-coded measures. Using a $t$-test, the correlations between the open text-based measures and the human-coded measures were compared with the average correlation between the semi-open text-based and the human-coded measures. To test hypotheses 1b and 1c, statistical differences in convergence between the separate text-based measures and the human-coded measures were assessed by means of Steiger's [94] $z$. Algorithm error and the corresponding hypothesis 2 were assessed by computing Krippendorff's α among the text-based measures. Transient error was examined by correlating the text-based measure from wave 1 with the text-based measure from wave 2. The test of hypothesis 3 was based on the comparison of the test-retest reliability of the text-based measures with the test-retest reliability of the human-coded measures (H3a) and with the test-retest reliability of the general closed job satisfaction measure (H3b).

Moving on to construct validity, hypothesis 4a was tested by computing the correlations between the text-based measures and the closed job satisfaction measures. Hypothesis 4b was assessed by $z$-transforming all correlations and comparing the correlation between text-based measures and the general job satisfaction measures with the correlations between the text-based measures and the job facet satisfaction measures by means of a $t$-test. Hypothesis 5a was tested by examining the correlations between the job satisfaction measures and the measures

**Table 4. Correlations between open text-based measures and closed job satisfaction question (N = 1,113) and test-retest reliability (N = 116).**

|  | Human coding | LIWC 2015 | SentimentR | SentiStrength | Closed question |
|---|---|---|---|---|---|
| Human coding | .543 |  |  |  |  |
| LIWC 2015 | .508 | .249 |  |  |  |
| SentimentR | .532 | .512 | .329 |  |  |
| SentiStrength | .587 | .510 | .487 | .189 |  |
| Closed question | .726 | .393 | .407 | .464 | .502 |

Test-retest reliability values are displayed on the diagonal; All correlations significant at the level of $p < .05$; LIWC = Linguistic Inquiry and Word Count

of the related constructs. Hypothesis 5b concerned comparisons between two sets of correlations, i.e. correlations between text-based measures and closed job satisfaction measure vs. correlations between text-based measures and measures of related but distinct constructs. We used Steiger $z$ tests to assess whether these differences were significant. Hypothesis 6a was supported if the closed question measures of job satisfaction correlated more strongly with the semi-open text-based measures than with the open text-based measures. We tested this hypothesis by $z$-transforming all correlations and comparing them using a $t$-test. Hypothesis 6b would be supported if the semi-open text-based measure' average deviation with measures of the other constructs was higher than the average deviation of the open text-based measures' deviation with these measures. The correlation analyses used for testing hypothesis 4 to 6 were based on the data from wave 1, as the job facet satisfaction questions and measures of the related constructs were not administered in wave 2.

# Results

## Measurement error

The text-based measures suffered from specific factor error to different degrees, as demonstrated in a wide range of parallel forms reliability values ($r_{min} = .189$ to $r_{max} = .775$). The correlations between the open text-based measures and the human-coded measure were lower ($r_{average} = .508$) than the correlations between the semi-open text-based and the human-coded measure ($r_{average} = .774$), as shown in Tables 4 and 5. We accepted hypothesis 1a, because this difference was statistically significant ($t = .563$, $p < .01$). As hypothesized, in the analysis of the responses to the open questions, the SentiStrength measure ($r = .587$) appeared to suffer less from specific factor error than the LIWC 2015 measure ($r = .508$, $t = 3.38$, $p < .01$) and the SentimentR measure ($r = .532$, $t = 2.33$, $p < .05$). Contrary to our expectations, the SentimentR measure did not suffer less from specific factor error than the LIWC 2015 measure ($t = 1.00$, n. s.). Therefore, we only partially accepted hypothesis 1b. With respect to the semi-open question, the results showed that the SentiStrength measure ($r = .695$) suffered more from specific

**Table 5. Correlations between semi-open text-based measures and closed job satisfaction question (N = 1,113) and test-retest reliability (N = 116).**

|  | Human coding | LIWC 2015 | SentimentR | SentiStrength | Closed question |
|---|---|---|---|---|---|
| Human coding | .314 |  |  |  |  |
| LIWC 2015 | .775 | .311 |  |  |  |
| SentimentR | .772 | .708 | .244 |  |  |
| SentiStrength | .696 | .704 | .665 | .250 |  |
| Closed question | .628 | .576 | .593 | .547 | .502 |

Test-retest reliability values are displayed on the diagonal; All correlations significant at the level of $p < .05$; LIWC = Linguistic Inquiry and Word Count

factor error that the LIWC 2015 measure ($r = .775$, $t = 5.63$, $p < .01$) and the SentimentR measure ($r = .772$, $t = 5.21$, $p < .01$). In addition, our results suggested that the specific factor error in the LIWC 2015 measure and SentimentR measure are equivalent ($t = 0.16$, n.s.). These findings provided only partial support for hypothesis 1c.

Algorithm error was high, as Krippendorff's $\alpha$ was generally low, i.e., $\alpha < .65$, confirming hypothesis 2. Notably, the algorithm error was lower for the semi-open text-based measures, i.e., $\alpha_{\text{all CATA}} = .506$, $\alpha_{\text{LIWC2015-SentimentR}} = .471$, $\alpha_{\text{LIWC2015-SentiStrength}} = .495$ and $\alpha_{\text{SentimentR-SentiStrength}} = .631$) than for the open text-based measures (i.e., $\alpha_{\text{all CATA}} = .187$, $\alpha_{\text{LIWC2015-SentimentR}} = .010$, $\alpha_{\text{LIWC2015-SentiStrength}} = .159$. and $\alpha_{\text{SentimentR-SentiStrength}} = .434$).

In our test of transient error, we discovered that the test-retest reliability of the open text-based measures (average $r_{tt} = .255$) was more than .2 lower than the human-coded measure ($r_{tt} = .543$). The test-retest reliability of the semi-open text-based measure (average $r_{tt} = .268$) deviated less than .2 from the test-retest reliability of its corresponding human-coded measure ($r_{tt} = .314$). In accordance, we could only partially accepted hypothesis 3a. The test-retest reliabilities of the open and semi-open text-based measures diverged substantially from the test-retest reliability of the measure based on the closed job satisfaction question. As a result, we rejected hypothesis 3b.

### Construct validity of textual job satisfaction measures

To test convergent validity, we correlated the text-based measures with the closed job satisfaction question measures. These findings are presented in Table 6. We found support for hypothesis 4a, because all text-based measures positively correlated with the closed questions ($r_{\min} = .203$; $r_{\max} = .579$). Hypothesis 4b was also supported, as an independent $t$-test showed that the correlations between the text-based measures and the general job satisfaction measure ($r_{\text{average}} = .532$) were consistently higher than the correlations between the text-based measures and the job facet satisfaction measures ($r_{\text{average}} = .356$; $t = 3.45$, $p < .05$).

We tested discriminant validity by correlating the text-based measures with closed question measures of P-O fit, virtuous leadership, life satisfaction, flourishing and OCB. As showed in Table 6, all correlations except for one (i.e., OCB–open text-based measure$_{\text{LIWC2015}}$) were

**Table 6. Correlations between text-based measures and closed question measures (N = 997).**

| Text-based measure | | General job satisfaction | Job facet satisfaction | | | | | | | Antecedents | | Outcomes | | |
|---|---|---|---|---|---|---|---|---|---|---|---|---|---|---|
| | | | Work environment | Work content | Team | Supervisor | Work-balance | Company | Pay | P-O fit | Virtuous leadership | Life satisfaction | Flourishing | OCB |
| Independent coder ratings | Open | .703 | .503 | .598 | .445 | .512 | .412 | .582 | .355 | .539 | .482 | .413 | .400 | .168 |
| | Semi-open | .618 | .458 | .555 | .361 | .438 | .370 | .537 | .336 | .487 | .427 | .361 | .326 | .128 |
| LIWC 2015 | Open | .373 | .303 | .333 | .230 | .244 | .233 | .337 | .200 | .290 | .234 | .233 | .233 | *.048* |
| | Semi-open | .564 | .412 | .497 | .329 | .376 | .309 | .474 | .301 | .443 | .359 | .318 | .288 | .092 |
| SentimentR | Open | .382 | .301 | .345 | .256 | .305 | .245 | .342 | .229 | .279 | .292 | .220 | .194 | .078 |
| | Semi-open | .579 | .444 | .511 | .367 | .423 | .328 | .504 | .277 | .440 | .397 | .295 | .279 | .074 |
| SentiStrength | Open | .457 | .374 | .396 | .288 | .300 | .286 | .368 | .199 | .360 | .289 | .287 | .283 | .074 |
| | Semi-open | .541 | .402 | .480 | .348 | .368 | .288 | .475 | .282 | .422 | .359 | .302 | .272 | .070 |

P-O = Person-Organization; OCB = Organizational citizenship behavior; LIWC = Linguistic Inquiry and Word Count, *N* = Sample size; All *p*s < .05, except for the one in italic.

positive and significant. Considering the evidence that the LIWC measure is much less reliable than the other CATA measures, we accepted hypothesis 5a. We also found support for hypothesis 5b, as the correlations between the text-based measures and the closed question measure of job satisfaction were consistently higher than the correlations between the text-based measures and the measures of the other constructs. As an illustration, the LIWC 2015 measure based on the open question was the most at risk for poor discriminant validity, as its correlation with the closed question job satisfaction measure diverged the least from its correlations with other constructs ($\Delta r$ = .083). Yet, a Steiger $z$ test showed that this difference was still significant ($t$ = 3.22, $p < .01$). The semi-open text-based measure had better convergent validity, as its correlation with the closed question measure of job satisfaction ($r_{average}$ = .510) was higher than the correlation between the open text-based measure and the closed question job satisfaction measure ($r_{average}$ = .457). This difference was statistically significant ($t$ = 5.70, $p < .05$). Similarly, the semi-open text-based measure correlated significantly stronger with the job facet questions ($r_{average}$ = .413) than the open text-based measure ($r_{average}$ = .300; $t$ = 4.39, $p < .01$). Hence, we accepted hypothesis 6a. The semi-open text-based measure also displayed better discriminant validity, as its average deviation with measures of the other constructs ($\Delta r_{average}$ = .267) was almost .09 higher than the average deviation of the open text-based measures' deviation with these measures ($\Delta r_{average}$ = .178). Hence, we found support for hypothesis 6b.

We conducted a robustness check that assess whet her the reliability and validity of CATA measures differed across lowly educated (2-year degree or lower) and highly educated respondents (4-year degree or higher). A comparison of correlations showed that parallel-forms reliability was .05 higher and convergent validity was .06 higher for the lowly educated respondents. These differences were not significant though ($t$ = .77 and $t$ = 1.12, respectively, $p$ = n.s.).

## Discussion

In recent years, CATA is being used increasingly often within and outside the organizational sciences [3]. In the case of sentiment analysis, most studies have created measures based on collections of lengthy texts. Consequently, computer-aided sentiment analysis has rarely been used to construct measures from responses to open survey questions, while such questions can be an informative complement to closed survey measures. In our study, we have started to fill this gap by demonstrating the reliability of lexicon-based sentiment analysis methods for constructing text-based job satisfaction measures and looking at their validity. We tested our hypotheses on cross-sectional data from 997 workers in the US and the UK and longitudinal data from 116 workers. In particular, we constructed text-based measures open and semi-open job satisfaction questions using three CATA techniques, LIWC 2015, SentimentR and SentiStrength, and a human coding procedure. As expected, measure construction by CATA methods took a negligible amount of the time (i.e., under half a minute). In sharp contrast, three manual coders required about 66 hours to annotate all texts. Next, we investigated the degree of measurement error in the different text-based measures (specific factor error, algorithm error and transient error) and examined their convergent and discriminant validity.

Concerning reliability, we demonstrated that specific factor error, the degree of convergence between the measures produced by CATA and the human coders, was lowest for the semi-open text-based measures and parallel forms reliability varied substantially across software packages. Algorithm error, the degree of agreement between text-based measures, was generally low. This lack of agreement is likely to be related to our decision to recode the software programs' original sentiment ratings into comparable 5-point Likert scales. This decision was problematic for our LIWC 2015 measure, as the recoding of its 1 to 100 scale resulted in a distribution of only very negative, moderately negative scores and very positive scores (Fig 1).

As the SentiStrength and SentimentR software produced ratings that were largely neutral or moderately positive, their agreement with the LIWC 2015 measures was very low and attenuated the average algorithm error. The transient error of the text-based measures was mostly in line with the transient error in the human-coded text-based measure, but consistently diverged from the transient error in the closed question job satisfaction measure.

Our initial test of construct validity showed that the open and, in particular, semi-open text-based measures have satisfactory convergent and discriminant validity. We found that the text-based measures correlated more strongly with closed question measures of general job satisfaction and job facet satisfaction and diverged more strongly from related but distinct constructs than the text-based measure based on the open question. This finding can be interpreted in various ways. If we assume that closed questions are the most suitable instrument for quantifying job satisfaction and consider the greater convergence and divergence of the semi-open text-based measure over the open text-based measure, we could argue that semi-open questions should be preferred for measuring job satisfaction. Alternatively, if we assume that closed job satisfaction questions inevitably fail to measure the construct in its entirety, the lack of convergence and divergence between the open text-based measure and closed job satisfaction measures can also refer to the complementary nature of open questions. Perhaps, the responses to open questions contain information about job characteristics that are not measured by closed questions.

## Limitations and future research

While our context-free sentiment dictionaries already produced reasonably reliable measures, future research would benefit from employing deductive and inductive dictionary-generation techniques to create a job satisfaction specific dictionary and thereby further boost reliability and, in turn, validity [4]. For example, researchers could look beyond unigrams, i.e., single words, and study the added value of multigrams, sequences of adjacent words [44]. Using the data from this study, researchers may discover that some words have different meanings in different contexts, e.g., the word 'challenging' may have very different connotations when it is used in combinations with words such as 'gratifying', 'motivating' and 'engaging' than with words such as 'busy', 'stressful' and 'exhausting'. Furthermore, scholars could explore the added value of learning-based sentiment analysis methods (see [95] for practical text mining guidelines), for example by training algorithms on our reliably labeled textual data. We note that high quality training data is costly to attain, as it usually involves tasking multiple coders to annotate texts. Survey researchers could ask respondents to rate their own textual responses in terms of sentiment toward the end of online surveys to have a reliable and time-saving alternative to a manual coding procedure. After all, respondents' own perceptions of sentiment are likely to come closest to the 'true', measurement-error-free sentiment score.

Our validation procedure suffered from several limitations. In our assessment of convergent validity, we, for example, did not examine the text-based measures' convergence with validated multiple-item job satisfaction scales or control for same-source variance. Therefore, we recommend future researchers to conduct an even more systematic validation of the new measures. The validation approach from Fisher, Matthews and Gibbons [96] could be followed, because the open and semi-open questions are single-item measures. In addition, future research could investigate whether the choice to produce text-based measures by means of sentiment analysis causes the measures to be more affect-oriented than cognition-oriented [30,97]. Scholars could test this by correlating the text-based measures with a closed question measure of job affect and a measure of job cognition [98]. Our examination of discriminant validity was limited, as the selection of constructs was small, all constructs were measured at

one point in time and all measures were self-report. Future studies could look into the text-based measures' relationships with a wider range of antecedents, objective outcome variables such as sickness absence and turnover, and supervisor-rated performance constructs such as productivity and creativity. In light of this, it could prove useful to assess the incremental validity of the text-based measures over the closed question measures.

The initial evidence from our study has opened interesting research venues for mixed method research. Open and, in particular, semi-open questions show great promise for measuring job satisfaction, because textual responses can reliably and swiftly be translated into text-based measures of job satisfaction, exhibit substantial convergence with closed question measures and display significant divergence with closed question measures of related but distinct constructs. We stress that (semi-)open questions should not just be regarded as another method to quantitatively measure a psychological construct. The information richness of the responses to open questions and semi-open questions can help scholars to unravel new insights about the sources and context of constructs. Whether used for cross-validation, contextualization or both, we believe that (semi-)open questions have the potential to further the science and practice of measuring and theorizing about psychological constructs.

## Supporting information

**S1 Protocol. Sentiment coding protocol.** (titled [CodingProtocol.pdf]).
(DOCX)

**S1 File. The R script used for data analyses.** (titled [PromiseOfOpenQuestions _Rscript. Rmd]).
(RMD)

**S1 Dataset. All data files used for this study.** (titled [Data_PromiseOfOpenQuestions.zip]).
(ZIP)

## Author Contributions

**Conceptualization:** Indy Wijngaards, Martijn Burger, Job van Exel.

**Data curation:** Indy Wijngaards.

**Formal analysis:** Indy Wijngaards.

**Funding acquisition:** Martijn Burger.

**Investigation:** Indy Wijngaards.

**Methodology:** Indy Wijngaards.

**Project administration:** Martijn Burger, Job van Exel.

**Supervision:** Martijn Burger, Job van Exel.

**Validation:** Indy Wijngaards, Martijn Burger.

**Visualization:** Indy Wijngaards.

**Writing – original draft:** Indy Wijngaards.

**Writing – review & editing:** Indy Wijngaards, Martijn Burger, Job van Exel.

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
