## [Decision Letter · Decision Letter 0]

24 Sep 2019

PONE-D-19-21718

The promise of open survey questions - The validation of text-based job satisfaction measures

PLOS ONE

Dear Mr. Wijngaards,

Thank you for submitting your manuscript to PLOS ONE. After careful consideration, we feel that it has merit but does not fully meet PLOS ONE’s publication criteria as it currently stands. Therefore, we invite you to submit a revised version of the manuscript that addresses the points raised during the review process.

We would appreciate receiving your revised manuscript by Nov 08 2019 11:59PM. To enhance the reproducibility of your results, we recommend that if applicable you deposit your laboratory protocols in protocols.io, where a protocol can be assigned its own identifier (DOI) such that it can be cited independently in the future. For instructions see: http://journals.plos.org/plosone/s/submission-guidelines#loc-laboratory-protocols

We look forward to receiving your revised manuscript.

Kind regards,

Irena Spasić

Academic Editor

PLOS ONE

Journal Requirements:

2. Please include additional information regarding the survey or questionnaire used in the study and ensure that you have provided sufficient details that others could replicate the analyses. For instance, if you developed a questionnaire as part of this study and it is not under a copyright more restrictive than CC-BY, please include a copy, in both the original language and English, as Supporting Information

Additional Editor Comments (if provided):

The related work on the use of text mining for survey data mining needs to be expanded. The following references need to be included:

Li, Hang, and Kenji Yamanishi. "Mining from open answers in questionnaire data." Proceedings of the seventh ACM SIGKDD international conference on Knowledge discovery and data mining. ACM, 2001. https://dl.acm.org/citation.cfm?id=502512.502579 also available in a journal at DOI: 10.1109/MIS.2002.1039833

Spasić, Irena, David Owen, Andrew Smith, and Kate Button. "Closing in on open–ended patient questionnaires with text mining." In Proceedings of the UK Healthcare Text Analytics Conference (HealTAC), Manchester, UK. 2018. http://orca.cf.ac.uk/109733/

Moreo, Alejandro, Andrea Esuli, and Fabrizio Sebastiani. "Building automated survey coders via interactive machine learning." International Journal of Market Research, 2019. https://journals.sagepub.com/doi/full/10.1177/1470785318824244

Reviewers' comments:

Reviewer's Responses to Questions

5. Review Comments to the Author

Reviewer #1: This article reports a study that compares measures of job satisfaction based on different methods of analysis of open questions and closed questions. The paper largely focuses on the nature of the methods and presents little evidence showing how the two approaches improve our understanding of the area. Construct validity would have been desirable, covering not just measurement of job satisfaction but job characteristics that influence it, and the outcomes following changes in satisfaction.

Reviewer #2: This paper presents a comparison of the reliability and validity of computer-aided text analysis techniques, comparing multiple forms of error associated with each, and examining their convergence with a closed-question concerned with job satisfaction. In addition to being innovative, it is an interesting and well-written paper that presents much new information that is quite useful. The fact that it is hypothesis-driven makes it, to me, even a superior effort. I thus really only have a few comments that I would ask the authors to address:

• Wondering if any respondent characteristics might be associated with the reliability or validity of the various approaches examined? For example, might the open-ended responses of older persons or less-educated persons be less reliable and/or valid? The data would seem to be available to investigate this.

• On page 15 of 35, a comparison of number of words per response at wave 1 vs. wave 2 is presented. More useful, though, might be a comparison of wave 1-wave 2 responses for those who completed both waves.

• There are several typos throughout the paper that will need to be corrected.

6. PLOS authors have the option to publish the peer review history of their article (what does this mean?). If published, this will include your full peer review and any attached files.

Reviewer #1: No

Reviewer #2: Yes: Timothy P Johnson

**Comments to the Author**

1. Is the manuscript technically sound, and do the data support the conclusions?

Reviewer #2: Yes

2. Has the statistical analysis been performed appropriately and rigorously? 

Reviewer #2: Yes

3. Have the authors made all data underlying the findings in their manuscript fully available?

Reviewer #2: Yes

4. Is the manuscript presented in an intelligible fashion and written in standard English?

Reviewer #2: Yes

---

## [Author Response · Author response to Decision Letter 0]

23 Oct 2019

Response to the editor:

*Thank you for your recommendations. We added the references to the manuscript as well as three related ones in the last sentence of the first paragraph of the Introduction. We used the references to substantiate the following claim “researchers outside the discipline have proposed a multitude of promising solutions for mining textual survey data”. We decided not to devote a full paragraph to related works on the broader discipline of survey data mining, as sentiment analysis rather than topic modelling is the subject of our paper, our paper has a relatively narrow scope and addressing the suggestions by the reviewers already lengthens the paper substantially. 

Response to reviewer 1:

We thank Reviewer 1 for the valuable comments. 

We agree that the paper primarily focuses on the measurement (reliability and validity) and lacks a thorough discussion on the ways it adds to the literature. We believe that this is an important research topic and recommend it for future research in our revised manuscript. We emphasized the importance in the end of the discussion section: “We stress that open questions should not just be regarded as another method to quantitatively measure a psychological construct. The information richness of the responses to open questions and semi-open questions can help scholars to unravel new insights about the sources and context of constructs.” We did not investigate the contextualizing potential op (semi-)open questions it in the current manuscript, because adding it would force the paper to be less focused and lengthen it too much. 

We followed Reviewer 1’s advice to take construct validity more seriously. We added a test of discriminant validity to the revised manuscript. We looked into the bivariate relationship of the textual measures with two antecedents, person-organization fit and virtuous leadership, and three outcome variables, organizational citizenship behavior, life satisfaction and flourishing. While this investigation further substantiates our conclusions about construct validity, we acknowledge that this selection of related but distinct constructs is limited, and validity can be examined in other ways. For this reason, we have proposed several additional research questions that can be addressed by future researchers (e.g., on incremental validity of the textual measures). 

Response to reviewer 2:

We thank Reviewer 2 for the kind words and valuable suggestions. We addressed all comments in the revised manuscript.

1. We added a few sentences on the effects of respondent characteristics on validity and reliability of textual measures in the Results section. We did not devote a hypothesis and section on the topic, as the most influential characteristic (presumably) was no significant predictor of reliability or validity. We did not include more comparative analyses (e.g., age), as it would lengthen the text substantially and would distract from the main topic of the paper. 

We examined whether education level was related to the specific factor error and convergent validity of the various text-based measures. We examined differences for lowly educated (i.e., 2-year degree or lower) vs. highly educated respondents (i.e., 4-year degree or higher). We could not formulate a single hypothesis, as we deemed two competing hypotheses defensible. One the one hand, lowly educated respondents may experience more troubles translating their feelings and cognitions in text than higher educated respondents, in turn making it more difficult for a CATA method to infer their true sentiment. As such, we could hypothesize that education level is positively related to reliability and convergent validity. On the other hand, having a relatively poorly developed vocabulary and experiencing feelings of discomfort when writing up texts may in fact boost the reliability and, in turn, validity of CATA measures. It is plausible that lowly educated respondents write more concise and semantically basic sentences and use less complex grammatical structures in their responses to open questions. Also, it would not be surprising that lowly educated respondents write down more straightforward adjectives. When we consider that complexity of texts and the reliability of CATA methods are negatively related, we could also hypothesize that education level is negatively related to reliability and convergent validity. The hypotheses about specific factor error and convergent validity were tested by first Fisher z transforming all correlations and then comparing them by means of a t-test.

We tested these hypotheses based on the existing data. As shown in the Table 1 (see Response to Reviewers), the results indicated that the reliability and convergent validity was higher for the sample of lowly educated respondents. Compared to the CATA measure based on the data from the highly educated sample, the CATA measures based on the lowly educated sample correlated about .05 more strongly with the human measure and about .06 more strongly with the closed question measure of job satisfaction. However, these differences were not significant (t = .77 and t = 1.12, respectively, p = n.s.).

2. We added this comparison in the revised manuscript next to the existing comparison. 

3. We, the three authors that wrote the paper, went over the text and attempted to correct all typos.

---

## [Decision Letter · Decision Letter 1]

27 Nov 2019

The promise of open survey questions - The validation of text-based job satisfaction measures

PONE-D-19-21718R1

Dear Dr. Wijngaards,

We are pleased to inform you that your manuscript has been judged scientifically suitable for publication and will be formally accepted for publication once it complies with all outstanding technical requirements.

With kind regards,

Irena Spasić

Academic Editor

PLOS ONE

Additional Editor Comments (optional):

Reviewers' comments:

Reviewer's Responses to Questions

**Comments to the Author**

1. If the authors have adequately addressed your comments raised in a previous round of review and you feel that this manuscript is now acceptable for publication, you may indicate that here to bypass the “Comments to the Author” section, enter your conflict of interest statement in the “Confidential to Editor” section, and submit your "Accept" recommendation.

Reviewer #2: All comments have been addressed

2. Is the manuscript technically sound, and do the data support the conclusions?

Reviewer #2: Yes

3. Has the statistical analysis been performed appropriately and rigorously? 

Reviewer #2: Yes

4. Have the authors made all data underlying the findings in their manuscript fully available?

Reviewer #2: Yes

5. Is the manuscript presented in an intelligible fashion and written in standard English?

Reviewer #2: Yes

6. Review Comments to the Author

Reviewer #2: thank you for this revised version. good work and appreciated reading it again. no further comments on this strong paper.

7. PLOS authors have the option to publish the peer review history of their article (what does this mean?). If published, this will include your full peer review and any attached files.

Reviewer #2: Yes: Timothy P Johnson

---

## [Editor Report · Acceptance letter]

6 Dec 2019

PONE-D-19-21718R1 

The promise of open survey questions - The validation of text-based job satisfaction measures 

Dear Dr. Wijngaards:

I am pleased to inform you that your manuscript has been deemed suitable for publication in PLOS ONE. Congratulations! Your manuscript is now with our production department. 

With kind regards,

on behalf of

Professor Irena Spasić 

Academic Editor

PLOS ONE